

# The Role of Response Efficacy and Self-efficacy in Disaster Preparedness Actions for Vulnerable Households

Dong Qiu 1, Binglin Lv 1, Yuepeng Cui1,*, and Zexiong Zhan1

1.School of Management, Fujian University of Technology, Fuzhou 350118, China; qiudong@fjut.edu.cn (D.Q.); lvbinglin@163.com (B.L.); chan@fjut.edu.cn(Z.Z)

*Correspondence to: ypcui916@hotmail.com;

**Abstract:** The effects of response efficacy and self-efficacy on disaster preparedness have been widely reported. However, most studies only prove their relationship to disaster preparedness in general terms without ascertaining whether they also variously impact the disaster preparedness of diverse vulnerable families (i.e., caring for older/disabled adults(COD), caring

for a child(CC), and low capacity (LC)). In this study, disaster preparedness is divided into two dependent variables: adequate and minimal preparedness. A quantitative analysis was conducted using 4559 samples drawn from the 2021 National Household Survey to investigate the relationship between response efficacy and self-efficacy with preparedness measures adopted by vulnerable households. Binary logistic regression results indicated that households with vulnerable groups are generally more likely to report lower disaster preparedness. Response efficacy is more critical to LC and COD

families, while self-efficacy is more important to LC and CC families. Based on these findings, interventions can be tailored to suit different family types and help vulnerable families better prepare for disasters.

**Keywords:** Disaster Preparedness, Response Efficacy, Self-efficacy, Vulnerable Households

## 1 Introduction

Global climate change has led to floods, droughts, tsunamis, wildfires, thunderstorms, hurricanes, etc., and the possibility of climate and weather disasters has increased exponentially (Rao et al., 2022). According to a survey by the United Nations Office for Disaster Risk Reduction, in the ten years from 2005 to 2014 alone, about 1.7 billion people the world over were affected by various disasters, among which the Chinese people suffered the most disasters, while the United

States sustained the most losses (United Nations Office for Disaster Risk Reduction, 2015). Disasters pose a considerable threat to human habitations and communities, often resulting in casualties besides various social, psychological, economic, or environmental losses (International Federation of Red Cross and Red Crescent Societies, 2020). However, most fatalities, destruction, and losses caused by disasters are preventable, so proactive measures are necessary to prepare people for these situations(Levac et al., 2012). Disaster preparedness refers to a state of readiness engendered by undertaking various

activities and employing resources. Individuals, communities, and organizations can effectively predict, respond, and recover from the effects of a disaster(United Nations International Strategy for Disaster Reduction, 2009). Adequate emergency



preparedness of households is one of the most effective ways of mitigating the impact of disasters (Gargano et al., 2015; Keim, 2008), and previous studies unmistakably confirm that the emergency preparedness of households significantly reduces the negative effects of disasters and ensures that people adequately support themselves and their families within 72 h

after the occurrence of a disaster (Levac et al., 2012; Malmin, 2021; Rao et al., 2022). For example, Cong et al. (2014) found that developing household emergency preparedness plans helps individuals adopt appropriate protective actions during tornadoes, thus reducing the number of injuries or deaths. Furthermore, people who prepare their households for emergencies report fewer worries and fears, which ensures they remain calm and act decisively to increase their chances of survival during natural disasters (Diekman et al., 2007).

However, a significant knowledge gap exists in this area of study, namely lack of research on enhancing the disaster preparedness capacity of families with particularly vulnerable groups. For example, children, the elderly, and people with disabilities or low mobility due to health conditions are generally considered particularly vulnerable groups. They often face differentiated challenges in emergencies and are also prone to be overlooked in the formulation of emergency preparedness plans and interventions (Fox et al., 2007; Smith and Notaro, 2009). Although the Theory of Planned Behavior (Martins et al.,

2019), Protection Motivation Theory (PMT) (Floyd et al., 2000a; Grothmann and Reusswig, 2006; Maddux and Rogers, 1983), Protection Action Decision Model (PADM) (Lindell and Perry, 2012), Social Learning Model (O'Brien et al., 2010), Health Belief Model (Rostami-Moez et al., 2020), Social Capital Framework (Koh and Cadigan, 2008; Martins et al., 2019), and numerous other conceptual frameworks are frequently used in disaster preparedness research, few empirical studies have covered the problems faced by these particularly vulnerable families. In addition, disaster preparedness aims to ensure

personal safety first, with property safety being secondary, so not all preparedness items are equally important. A lack of flashlights, for example, is less detrimental to a family than a lack of food or an evacuation vehicle. Therefore, following Malmin (2021) and Rao (2022), we too subdivide disaster preparedness into minimal preparedness (i.e., items that ensure the safety of life, such as water, food, and transportation) and adequate preparedness (i.e., items that add finance, insurance, and other property safety to the minimum preparedness). Based on the theory of PMT and PADM, this study first analyzes the

factors influencing families to adopt adequate preparation and minimal preparedness, respectively. Then the moderating effects of response effectiveness and self-efficacy on disaster prevention behavior in vulnerable households were investigated, given the particular vulnerability of families with children, the elderly, the disabled, or members with reduced mobility due to health conditions. For representational purposes, we will use caring for older/disabled adults (COD), caring for a child (CC), and low capacity (LC), respectively, to represent families with elderly or disabled adults in need of care,

children in need of care, and families with members having reduced mobility due to disability or health conditions and without care services.

The main contributions of this study to the field are as follows: 1) Family-based disaster prevention work has yet to be thoroughly studied. This paper fills several research gaps in family disaster preparedness by exhaustively describing the key influencing elements of family preparedness. 2) Response efficacy and self-efficacy display differentiated mediate effects

based on the composition of vulnerable families. 3) Response efficacy and self-efficacy mediate only the minimal



preparedness action for vulnerable families and fail to motivate such families to be adequately prepared. Besides, this study provides direction for further research, enabling vulnerable families to transition from minimal preparation to adequate preparation.

The rest of this paper is arranged as follows: the second part reviews relevant literature and puts forward the research
hypotheses; the third part introduces the research data and regression model; the fourth part analyzes the empirical results; the fifth part discusses the research findings; the sixth part comprises of the conclusion and suggestions.

## 2 Literature review and research hypotheses

### 2.1 Theoretical Framework

To explore disaster preparedness, this study blends the two conceptual frameworks of PMT and PADM. They consider the same objects but accord different priorities. The PMT, which was initially used to explain how fear affects individual and individual-related health adoption behaviors, emphasizes that protective motivation stems from individuals' desires to avoid
potential adverse outcomes after a perceived threat and is one of the most influential theories explaining individual risk prevention and adoption of protective actions (Rogers, 1975). It is now widely used to describe self-protective behaviors in the context of disasters (Bubeck et al., 2012; Grothmann and Reusswig, 2006; Floyd et al., 2000b). PMT believes that forming protective motivation (whether people adopt protective behaviors against potential threats) is a decision that people form after a combination of threat and response appraisal. Threat and response appraisal results are used to determine
whether a protective motivation exists that ultimately leads to a change in behavior. The theory suggests that the best measure of protection is behavioral willingness. Threat appraisal (i.e., perceived vulnerability and severity of a disaster) refers to people's awareness of the risk factors. Coping appraisal assesses an individual's ability to cope with and avoid risks and people's cognition of their ability to deal with health threats. Response appraisal evaluates an individual's ability to cope with and avoid risk. It is a perception of the health threat processing ability, including response efficacy (i.e., the perceived
effectiveness of disaster preparedness actions) and self-efficacy (i.e., the ability to perceive disaster preparedness actions). Similarly, PADM emphasizes that threat perception (i.e., perceived consequences of a disaster), protective action perception (i.e., effectiveness and cost of protective action), and stakeholder perception (i.e., stakeholders' power to mutually determine protective action) are crucial for taking protective action decisions (Lindell and Perry, 2012).

According to Grothmann (2006) and Poussin (2014), threat appraisal only appeared to contribute motivating energy for
a reaction; coping appraisal (i.e., self-efficacy and response efficacy) determined whether or not it was protective. Therefore, we focus our research on the concepts of "response efficacy" and "self-efficacy" within the protective motivation theory. Response efficacy refers to an individual's belief or perception regarding whether a protective behavior is effective. Generally, people adopt an action based on the belief that they would benefit from that action in a personally meaningful manner. Self-efficacy refers to an individual's belief or perception about her/his ability to adopt a specific protective





behavior, which is the belief, judgment, and subjective self-feeling that an individual has about the level at which s/he can complete the behavior before performing a particular behavior operation at the core of the protective motivation theory. Self-efficacy is crucial to the formation and change of behavior. The stronger the efficacy, the greater the possibility of behavior formation and change. In disaster research, self-efficacy assesses one's ability to initiate or complete a preventive, protective, or adaptive behavior. Response efficacy evaluates the utility and effectiveness of initiating or completing a preventive,

protective, or adaptive behavior. Previous studies indicate that response efficacy and self-efficacy can promote household disaster preparedness. For example, Wai et al. (2010) found that improved self-efficacy helped enhance disaster evacuation. Bubeck et al. (2013) investigated 752 flood-prone households along the Rhine River and opined that high self-efficacy and response efficacy could improve the possibility of these households adopting flood mitigation behaviors such as the deployment of flood barriers and the purchase of flood insurance. Further, Tang and Feng (2018) assert that self-efficacy and

response efficacy were significantly correlated with actual disaster preparedness behaviors. A survey by Lee and You (2020) during the early stages of COVID-19 in South Korea confirmed the importance of self-efficacy and response efficacy, which are related to behavioral responses and significantly influenced the level of preparedness of the people for the COVID-19 pandemic.

**2.2 Disaster Preparedness and Vulnerable Households**

      According to a recent report issued by the U.S. Census Bureau, about 32.5 million people have a severe disability, 63.1 million adults live with children under the age of 18, and 41.8 million adults become caregivers for an elderly family member who is ill or disabled. Not surprisingly, children, the elderly, and people with disabilities are highly dependent on

caregivers from their families who support them with meals, assistance in daily routines, and healthcare services. Due to their unique levels of physical, psychological, and behavioral development and their reliance on others to protect and care for them, children face greater risk in disasters. Similarly, older adults generally find coping with disasters more challenging due to their reduced mobility, health conditions, age-related decline in cognitive and sensory abilities, and a unique need for care and support. In contrast, people with disabilities usually experience the highest risk levels. Their reduced ability to hear and

understand disaster warnings due to physical, cognitive, or sensory impairments limits their ability to respond appropriately to disasters. To prevent or mitigate the negative impact disasters have on vulnerable groups, disaster preparedness for children, the elderly, or people with disabilities is undertaken by caregivers. Families with caregiving responsibilities are often perceived as contributing to family preparedness. For example, a study in Los Angeles reported that people with children under 18 at home were more likely to have an emergency plan for dealing with terrorism than those without

children. In addition, Chaney et al. (2013) indicated that families with children were more likely to have an emergency preparedness plan in place in advance than families without children. Household disaster preparedness is indispensable for families of children with disabilities, as their requirements could be more pronounced than those of families without disabled children (Hamann et al., 2016). Christensen (2012) studied the hurricane preparedness of caregivers of older adults with



cognitive disabilities in Florida and found that most home caregivers were well prepared. In Canada, family caregivers of
elderly stroke victims with functional disabilities were also found to be well-prepared to sustain themselves and their charges
for three days after a disaster. The main reason for families with caring responsibilities being better prepared is that they
anticipate the significant needs of those being cared for during a disaster.

### 2.3 Self-efficacy, Vulnerable Households, and Disaster Preparedness

Self-efficacy is a central component of many health behavior models and is generally understood to be an essential
factor in determining individual well-being and reducing vulnerability (Bandura, 2001; Lindell and Perry, 2012; Maddux and
Rogers, 1983; Rogers, 1975). Self-efficacy was found to enhance disaster preparedness in people with special needs and
vulnerabilities, such as children, the elderly, the disabled, people living with disabilities and health conditions, and those who
are underserved and under-resourced(Adams et al., 2017; Hamann et al., 2016; Marceron and Rohrbeck, 2019; Rivera, 2020;
Wirtz and Rohrbeck, 2018). Adams et al. (2017) found that self-efficacy could regulate the relationship between respondents'
health perceptions and their preparedness. Marceron and Rohrbeck (2019) believed that self-efficacy could increase the
probability of individuals adopting disaster preparedness measures by amplifying the efficacy of threat perception. Studies
have found no correlation between threat perception and disaster preparedness in individuals with low self-efficacy; in other
words, in the absence of self-efficacy, even a high level of threat perception had little impact on disaster preparedness.
Families with particularly vulnerable groups tend to have relatively low levels of self-efficacy. For example, in a comparison
of the self-efficacy of those with physical disabilities and non-physical disabilities in a study on persons with disabilities, it
was found that people with physical disabilities had lower self-efficacy in certain aspects (Becker and Schaller, 1995).
Similarly, in a study of patients with cerebral palsy, self-efficacy scores tended to be lower than average (Gaskin and Morris,
2008). Gist and Mitchell (1992) argued that certain interventions could alter the self-efficacy beliefs of vulnerable groups,
which in turn could be a means of improving their emergency preparedness (Bandura, 2001; Bandura et al., 1999). Hamann
et al. (2016) have also established that family self-efficacy corresponds to disaster preparedness levels, especially for
children with disabilities, and improving their self-efficacy in various activities can effectively enhance their disaster
preparedness abilities.

### 2.4 Response Efficacy, Vulnerable Households, and Disaster Preparedness

Response efficacy opens a window onto the cognitive processes underlying stimulus-preparation behavior and is a
strong predictor of current and future behavior (Milne et al., 2000). Many existing studies have explained the relationship
between response efficacy and disaster preparedness. In the United States, response efficacy was found to be an essential
factor in motivating homeowners to adopt steps to mitigate the effects of wildfires (Martin et al., 2007). Kerstholt et al.
(2017), in a survey of 629 citizens in The Hague, found that indicators such as response efficacy, community efficacy, trust,



and empowerment can directly or indirectly enhance citizens' flood control capacity. In a decision-making factor report on the Mauritians' willingness to pay to strengthen their epidemic preparedness, Jeetoo et al. (2022) stated that response

efficacy significantly impacted both the respondents' willingness to pay and the amount to be paid. Grothmann and Reusswig (2006) reported that response efficacy, self-efficacy, and costs were positively correlated to the four protective responses to the flooding protection afforded to German residents. Similarly, Tang and Feng (2018) found that after the earthquake in Taiwan, residents' sense of response efficiency could affect their actual disaster preparedness behavior by enhancing their willingness for disaster preparedness. Recent research by Chen and Cong (2022) revealed that response efficacy could

effectively enhance the level of disaster preparedness of respondents, especially for disasters with short lead times and for the elderly.

### 2.5 Hypotheses

Based on the above discussion, we developed four hypotheses; each hypothesis covers the three conceptualizations of preparedness.

**Hypothesis 1:** Families with stronger self-efficacy or response efficacy were more likely to be prepared for the disaster.

**Hypothesis 2:** Families with the three types of vulnerable groups (i.e., (1) families with lower capacity and (2) families with the elderly or disabled to care for (3) families with children to care for) will report better disaster preparedness than those

without.

**Hypothesis 3:** The moderating effects of response efficacy and self-efficacy differ with the composition of vulnerable families.

### 3 Methods

### 3.1 Data source

The data used in this study are drawn from the 2021 National Family Survey (NHS), a national survey administered by the Federal Emergency Administration (FEMA). The NHS is a nationally representative dataset that evaluates disaster preparedness in the United States over time, beginning in 2007, and is an annual exercise since 2013. In 2021, after random

numbers were drawn nationally, the survey was conducted simultaneously in English and Spanish and interviewed 7,197 adult respondents employing the telephone and the Internet. The survey included a nationwide representative sample and oversamples for specific hazards, such as drought, power outages, thunderstorms, tsunamis, volcanic eruptions, and winter storms. The NHS aims to track personal and family preparedness by investigating the public's preparedness behavior, attitudes, and motivation. The working sample of this study included 4,559 out of 7,197 respondents who provided valid

answers about disaster preparedness.



### 3.2 Measures

#### 3.2.1 Dependent variables

The dependent variables were measured using nine indicators regarding the multiple components of disaster preparedness in the NHS. The nine disaster preparedness indicators extracted from the questionnaire are given in Table 1. Referring to the analysis strategy employed by Malmin (2021) and Rao (2022), we classified preparedness into two separate measures, adequate and minimal preparedness. The nine disaster preparedness indicators are given in Table 1. Those whose response was None scored 0, and those who responded with Yes or other valid answers scored 1. Adequate preparedness was

measured as an indicator variable with a score of 1 if the respondent possessed at least 5 of the nine preparedness; otherwise, adequate preparedness was coded 0. The last measure was a newly created measure of minimal preparedness, a separate dichotomous measure that captured the essential elements necessary for immediate evacuation or sheltering in place for three days (Malmin, 2021; Rao et al., 2022). These most basic elements include essential supplies to get through, water, and access to transportation; respondents without all three of these items were coded as 0, and those who were prepared with one

of these minimal elements were coded as 1 (Malmin, 2021; Rao et al., 2022).

**Table 1 Disaster preparedness indicators**

| Indicators | Questions | Question Code |
|---|---|---|
| 1 | have adequate supplies for more than three days in an emergency | SUPP |
| 2 | have adequate water for more than three days in an emergency | RUNW |
| 3 | have transportation available for emergency evacuation | TRAN |
| 4 | have an emergency plan or some preparedness | POWE |
| 5 | have an emergency power supply | PREPB |
| 6 | Be aware of the Emergency Plan(s) of school(s), workplace(s), or community center(s) | EPSW |
| 7 | have money set aside for an emergency | FINR1 |
| 8 | have homeowners or renters' insurance for the residence | FP1 |
| 9 | have hazard-specific insurance | FP5 |

#### 220   3.2.2 Independent variables

We identified a CC family by querying, "How many household members are children under 18?" (0 = no, 1 = yes). COD families were ascertained through the question, "Do you currently live with or have primary responsibility for assisting an elderly person or someone with a disability who requires assistance (mobility, hearing, vision, cognitive, or intellectual disability or physical, mental, or health condition)?" (0 = no, 1 = yes). LC families were a dichotomous variable measured by




enquiring, "Do you have a disability or a health condition that might affect your capacity to respond to an emergency (a mobility, hearing, vision, cognitive, intellectual or physical disability, mental, or health condition)?" (0 = no, 1 = yes).

### 3.2.3 Moderators


Moderating variables are obtained through response efficacy and self-efficacy. Response efficacy is an individual's perception of preparedness actions that can effectively reduce risks, which was assessed by the question: "How much would taking steps to prepare to help you get through a disaster in your area?" and the answers ranged from 0 = not at all, 1 = Very little, 2 =somewhat, 3 = Quite a bit, to 4 = A great deal. Self-efficacy was measured by asking: "How confident are you that you can take steps to prepare for a disaster in your area?" The options ranged from 0 = not at all confident to 4 = extremely confident.

### 3.2.4 Control variables


Various factors influencing preparedness were controlled for in the study. Control variables included age, gender, race, ethnicity, homeownership, educational level, disaster information, number of adults, and household income. Age was coded as 0 = 18–59, 1 = Over 80. Gender was measured by 0 = male and 1 = female. We coded the variable Race as a dummy variable, with White being the reference group (0 = white, 1 = others). We assessed Ethnicity by asking, "Are you of Hispanic, Latino, or Spanish origin?" (0 = no, 1 = yes). Homeownership was dichotomously measured by 0 = renting a home and 1 = owning a home. Educational level was initially measured on a 6-point scale with 0 = Less than a high school diploma, 1 = High school degree or diploma, 2 = Some college, no degree, 3 = Associate's degree, 4 = Bachelor's degree, 5 = Post graduate work/degree or professional degree. Disaster Information access was assessed by asking, "A "disaster" is an event that could threaten lives, disrupt public or emergency services like water and power, or damage property. What information have you read, seen, or heard in the past year about how to better prepare for a disaster?" (0 = no, 1 = yes). The Number of Adults was a continuous variable, ranging from 1 to 11. Household Income was divided into nine grades, from 0 = Less than 59,999 to 1 = 60,000 or more. Prior Disaster Experience was measured by asking, "Have you or your family ever experienced the impacts of a disaster?" (0 = no, 1 = yes).

Based on the PMT & PADM, we considered a perceived risk to be an essential factor that affects disaster preparedness and hence controlled it in this study. Perceived risk was assessed with the question: "A "disaster" is an event that could threaten lives, disrupt public or emergency services like water and power, or damage property. Thinking about the area you live in, how likely was it for a disaster to impact you?" (0 = unlikely, 1 = very likely/likely).

### 3.3 Data analyses





Firstly, we calculated the descriptive statistics on adequate and minimum preparedness. Secondly, we analyzed the descriptive statistics of the independent and control variables. Finally, we used binary logistic regression to test the relationship between adequate and minimum preparedness and each element. Each of the two conceptualizations of preparedness includes seven models (i.e., 14 regression models in total). Model 1 only has independent variables and control variables. We added the bidirectional interaction between response efficacy and particularly vulnerable families in models 2–

4 to examine the moderating effect of response efficacy on core independent variables in two conceptualizations of preparedness. Models 5–7 amplify the two-way interaction between self-efficacy and particularly vulnerable families, besides testing the effects of self-efficacy on independent variables in two conceptualizations of preparedness. We performed all analyses using STATA 17.

**4 Results**

        Descriptive statistics describe the overall disaster preparedness and disaster-related independent variables as well as control variables. The sample included 4,559 respondents, of which 85.96% were considered adequately prepared and 11.12% were deemed as not meeting the minimum preparedness requirements. Figures for males (50.93%) were slightly

higher than those for females (49.07%), whites accounted for the most significant proportion of the sample (75.43%), English was the primary language in use in 92.10% of families, and the Hispanic population accounted for 19.76% of the sample respondents. In the sample, 26.50% of respondents were caregivers of older or disabled family members, relatives, or friends, 18.38% of the respondents only had high school level education or lower, and more than half the respondents had a bachelor's degree or higher education. Age was dichotomously measured as 0 = younger adults aged between 18 and 59 and

1 = older adults who were 60 or older. Of the respondents, 64.27% said they or their family had experienced a disaster in their lifetime, 67.43% of the respondents owned a home, and 29.68% of them had disabilities or health status. About 46.87% of the sample had an annual household income below $59,999, and 53.13% reported an annual household income over $60,000. Approximately 42.00% of the respondents reported being unemployed. More than 82% of the respondents believed disasters were likely to occur in their area. The respondents displayed relatively high levels of coping appraisal, with 29.85%

having moderate or high self-efficacy in their disaster preparedness capacity. Response efficacy was rated moderately higher at 61.06%. Nearly 96% of the respondents had received information on how to respond to a disaster in the past year.

**Table 2 Descriptive statistics for dependent variables (N = 4559)**

| Variables | N | Freq. | Percent |
|---|---|---|---|
| Adequate Preparedness (Measure of at least five of the nine items) | 4559 | | |
| 0 = No | | 640 | 14.04 |
| 1 = Yes | | 3,919 | 85.96 |



| | N | Freq. | Percentage |
|---|---|---|---|
| Minimal Preparedness (Separate measure of three most basic preparedness elements) | 4559 | | |
| 0 = No | | 507 | 11.12 |
| 1 = Yes | | 4,052 | 88.88 |

**Table 3 Descriptive statistics for independent variables and covariates (N = 4559)**

| Variables | N | Mean | Freq. | SD | Percentage |
|---|---|---|---|---|---|
| **Disaster Information** | 4559 | | | | |
| No | | | 187 | | 4.10 |
| Yes | | | 4,372 | | 95.90 |
| **Response efficacy** | 4559 | | | | |
| Not at all | | | 101 | | 2.22 |
| Very little | | | 508 | | 11.14 |
| Somewhat | | | 1,166 | | 25.58 |
| Quite a bit | | | 1,154 | | 25.31 |
| A great deal | | | 1,630 | | 35.75 |
| **Self-efficacy** | 4559 | | | | |
| Not at all confident | | | 105 | | 2.30 |
| Slightly confident | | | 543 | | 11.91 |
| Somewhat confident | | | 1,074 | | 23.56 |
| Moderately confident | | | 1,476 | | 32.38 |
| Extremely confident | | | 1,361 | | 29.85 |
| **Risk perception** | 4559 | | | | |
| Unlikely | | | 808 | | 17.72 |
| Likely | | | 3751 | | 82.28 |
| **Experience of Disaster** | 4559 | | | | |
| No | | | 1,629 | | 35.73 |
| Yes | | | 2,930 | | 64.27 |
| **Income** | 4559 | | | | |
| Less than 59,999 | | | 2,137 | | 46.87 |
| 60,000 or more | | | 2,422 | | 53.13 |
| **Caring for older/disabled adults** | 4559 | | | | |
| No | | | 3,3515 | | 73.50 |
| Yes | | | 1,208 | | 26.50 |
| **Age** | 4559 | | | | |
| 18–59 | | | 3,430 | | 75.24 |
| Over 60 | | | 1,129 | | 24.76 |





| | | | |
|---|---|---|---|
| **Gender** | 4559 | | |
| Male | | 2,322 | 50.93 |
| Female | | 2,237 | 49.07 |
| **Education** | 4559 | | |
| Less than a high school diploma | | 86 | 1.89 |
| High school degree or diploma | | 752 | 16.49 |
| Some colleges, no degree | | 890 | 19.52 |
| Associate's degree | | 504 | 11.06 |
| Bachelor's degree | | 1,121 | 24.59 |
| Post-graduate work/degree or professional degree | | 1,206 | 26.45 |
| **Race** | 4559 | | |
| White | | 3,439 | 75.43 |
| Others | | 1,120 | 24.57 |
| **Lower Capacity** | 4559 | | |
| No | | 3,206 | 70.32 |
| Yes | | 1,353 | 29.68 |
| **Homeownership** | 4559 | | |
| Rent | | 1,485 | 32.57 |
| Own | | 3,074 | 67.43 |
| **Employment** | 4559 | | |
| No | | 1,915 | 42.00 |
| Yes | | 2,644 | 58.00 |
| **Hispanic** | 4559 | | |
| No | | 3,658 | 80.24 |
| Yes | | 901 | 19.76 |
| **English** | 4559 | | |
| No | | 360 | 7.90 |
| Yes | | 4,199 | 92.10 |
| **Number of adults** | 4559 | 2.94 | 1.46 |
| **Caring for Children** | 4559 | | |
| No | | 2,522 | 55.32 |
| Yes | | 2,037 | 44.68 |

## 4.1 Adequate Preparedness

The binary logistic regression results on adequate preparedness on independent variables and covariates were
statistically significant (Table 5, P<0.001). Adequate preparedness measures whether a household has at least five of the nine



preparedness indicators. The independent variables LC and CC lacked statistically significant differences in terms of adequate preparation. Families with COD were 61% better prepared than those without (OR = 1.61, P <0.001). Families that received disaster preparedness information most recently were 5.16 times more likely to be well prepared than households that did not receive the same (OR = 5.16, P <0.001). Families that had experienced a disaster were 1.73 times more well-prepared than those that had not (OR = 1.73, P <0.001). Age and education levels were not significantly associated with the preparation, nor the number of adults in a household. Families that owned their homes were nearly three times more likely to be prepared than those that rented (OR = 2.76, P <0.001). Families with higher income levels were more likely to be prepared (OR = 1.72, P<0.001). Females were 27% less likely to be well prepared than males (OR = 0.73, P<0.05). Age, education levels, race, ethnicity, language, and the number of adults in the household were not significantly associated with adequate preparation. Families' levels of risk perception did not correspond to adequate disaster preparedness. Interestingly, self-efficacy was considerably correlated with adequate preparation (OR = 1.39, P< 0.001), while response efficacy and preparedness were not statistically significantly different. We added the interaction terms of reaction efficacy and self-efficacy with vulnerable families in models 2–7. The results revealed that none of the interaction terms were significant, indicating that improvement in response efficacy and self-efficacy did not always increase the probability of vulnerable families adopting adequate preparation.

**Table 4  Logistic regression results for rdequate preparedness**

| Variables | Adequate Preparedness | | | | | | |
|---|---|---|---|---|---|---|---|
| | Model 01 | Model 02 | Model 03 | Model 04 | Model 05 | Model 06 | Model 07 |
| Lower Capacity (ref =no)[a] | 1.09 | 2.01* | 1.09 | 1.09 | 1.08 | 1.09 | 1.09 |
| Caring for older/disabled adults (ref =no) | 1.61*** | 1.61*** | 2.24* | 1.61*** | 1.61*** | 1.14 | 1.61*** |
| Caring for Children (ref =no) | 1.03 | 1.02 | 1.02 | 1.65 | 1.03 | 1.03 | 1.05 |
| Age (ref = 18–59) | 1.07 | 1.07 | 1.08 | 1.08 | 1.07 | 1.08 | 1.07 |
| Gender (ref =male) | 0.73** | 0.73** | 0.73** | 0.73*** | 0.73** | 0.73*** | 0.73** |
| Education | 1.02 | 1.02 | 1.02 | 1.02 | 1.02 | 1.02 | 1.02 |
| Race (ref =white) | 0.98 | 0.97 | 0.98 | 0.98 | 0.98 | 0.98 | 0.98 |
| Hispanic (ref =no) | 1.16 | 1.14 | 1.15 | 1.15 | 1.16 | 1.16 | 1.16 |
| English (ref =no) | 1.24 | 1.24 | 1.23 | 1.24 | 1.24 | 1.24 | 1.24 |
| Income (ref =less than $59,999) | 1.72*** | 1.72*** | 1.71*** | 1.72*** | 1.72*** | 1.71*** | 1.72*** |
| Disaster Information (ref =no) | 5.16*** | 5.13*** | 5.13*** | 5.16*** | 5.16*** | 5.19*** | 5.15*** |
| Response efficacy | 1.07 | 1.13* | 1.09 | 1.13* | 1.07 | 1.07 | 1.07 |
| Self-efficacy | 1.39*** | 1.39*** | 1.39*** | 1.39*** | 1.39*** | 1.36*** | 1.39*** |





| | | | | | | | |
|---|---|---|---|---|---|---|---|
| Risk perception (ref =no) | 1.25 | 1.24 | 1.24 | 1.24 | 1.25 | 1.25 | 1.25 |
| Experience of disaster (ref =no) | 1.73*** | 1.72*** | 1.73*** | 1.72*** | 1.73*** | 1.73*** | 1.73*** |
| Homeownership (ref =rent) | 2.76*** | 2.74*** | 2.76*** | 2.75*** | 2.76*** | 2.76*** | 2.76*** |
| Employment (ref =no) | 1.27* | 1.27* | 1.27* | 1.27* | 1.27* | 1.27* | 1.27* |
| Number of adults | 0.99 | 0.99 | 0.99 | 0.99 | 0.99 | 0.99 | 0.99 |
| Lower Capacity ×Response efficacy | | 0.84 | | | | | |
| Caring for older/disabled adults×Response efficacy | | | 0.91 | | | | |
| Caring for Children×Response efficacy | | | | 0.88 | | | |
| Lower Capacity ×Self-efficacy | | | | | 1.00 | | |
| Caring for older/disabled adults×Self-efficacy | | | | | | 1.11 | |
| Caring for Children×Self-efficacy | | | | | | | 0.99 |
| **Model fit** | | | | | | | |
| LR Chi-square | 604.79 | 608.29 | 605.59 | 607.18 | 604.79 | 605.75 | 604.79 |
| Degree of freedom | 18 | 19 | 19 | 19 | 19 | 19 | 19 |
| p-value | | | | P<0.001 | | | |
| Pseudo R-squared | 16.35% | 16.45% | 16.37% | 16.42% | 16.35% | 16.38% | 16.35% |

[a]The reference categories are in parentheses

*p < .05, **p < .01, ***p < .001


### 4.2 Minimal Preparedness

The logistic regression for minimal preparedness on the key independent variables in this study was statistically
significant (Table 6, P<0.001). Minimal preparedness measures whether a family has three key elements – food, water, and a
vehicle. Families with care responsibilities had a lower rate of minimal preparedness than those without. The odds of
families with COD displaying minimal disaster preparedness were 33% lower than those without (OR = 0.67, P<0.01).
Families with children had nearly 62% lower odds of minimal preparedness (OR = 0.38, P<0.001). Respondents who
reported decreased capacity due to disability or health issues had a 26% lower chance of having minimal preparedness than
unreported respondents (OR = 0.74, P<0.05). The control variables of race, ethnicity, language, income, number of adults,
risk perception, and minimal preparedness were not significantly associated. Older respondents were 2.46 times more likely
to have minimal preparedness than younger respondents (OR = 2.46, P<0.001). Compared to males, females were 1.39 times
more likely to display minimal preparedness (OR = 1.39, P<0.01). Interestingly, families with higher levels of education
were 18 percent less likely to present minimal preparedness than those with lower levels of education (OR = 0.82, P<0.001).
Homeowners were 1.48 times more likely to be prepared than renters (OR = 1.48, P<0.001). Households affected by



disasters in the past were 1.35 times more likely to be prepared than those who were unaffected (OR = 1.35, P<0.01). Those who received preparedness information in the past year were 76% more likely to prepare for a disaster than those who did not receive similar information (OR = 1.76, P<0.05). Curiously, unemployed respondents were 26 percent more likely to be prepared than those who were employed (OR = 0.74, P<0.05). Analogous to adequate preparedness in moderating variables, we found self-efficacy to be significantly correlated with minimal preparedness, while there was no statistically significant

correlation between response efficacy and minimal preparedness.

    We added the interaction terms of the independent and moderator variables, respectively, in models 2–7. In models 2–3, we observed a significant interaction between response efficacy and LC (OR = 1.33, P<0.01) and COD (OR = 1.21, P<0.05) households, suggesting that a higher level of response efficiency could significantly improve the probability of minimal preparedness for LC and COD households. Model 4 demonstrates that the interaction term between response efficacy and

CC families is not statistically significant, meaning that response efficacy has no statistical correlation with the minimal preparedness of CC families. The interaction of self-efficacy with LC and CC households in models 5 and 7 was statistically significant, implying that self-efficacy could also significantly improve the likelihood of minimal preparedness for LC and CC households. The results for model 6 are insignificant, indicating that the improvement of self-efficacy does not necessarily increase the probability of COD households displaying minimal preparedness.


**Table 5 Logistic regression results for minimal preparedness**

| Variables | Minimal preparedness | | | | | | |
|---|---|---|---|---|---|---|---|
| | Model 01 | Model 02 | Model 03 | Model 04 | Model 05 | Model 06 | Model 07 |
| Lower Capacity (ref =no)[a] | 0.74* | 0.25*** | 0.74* | 0.73* | 0.36** | 0.73* | 0.74* |
| Caring for older/disabled adults (ref =no) | 0.67** | 0.67** | 0.32** | 0.67** | 0.66*** | 0.46* | 0.67** |
| Caring for Children (ref =no) | 0.38*** | 0.39*** | 0.39*** | 0.46* | 0.39*** | 0.38*** | 0.99 |
| Age (ref = 18–59) | 2.46*** | 2.51*** | 2.45*** | 2.47*** | 2.49*** | 2.47*** | 2.46*** |
| Gender (ref =male) | 1.39** | 1.38** | 1.38** | 1.38** | 1.37** | 1.38** | 1.38** |
| Education | 0.82*** | 0.82*** | 0.82*** | 0.82*** | 0.82*** | 0.82*** | 0.82*** |
| Race (ref =white) | 0.86 | 0.86 | 0.86 | 0.86 | 0.86 | 0.86 | 0.86 |
| Hispanic (ref =no) | 1.08 | 1.11 | 1.09 | 1.08 | 1.09 | 1.08 | 1.07 |
| English (ref =no) | 1.22 | 1.22 | 1.23 | 1.22 | 1.24 | 1.23 | 1.22 |
| Income (ref =less than $59,999) | 1.17 | 1.17 | 1.18 | 1.17 | 1.16 | 1.17 | 1.19 |
| Disaster Information (ref =no) | 1.76* | 1.86* | 1.85* | 1.74* | 1.84* | 1.80* | 1.64 |
| Response efficacy | 1.00 | 0.87* | 0.91 | 1.03 | 0.99 | 0.99 | 1.01 |
| Self-efficacy | 1.21*** | 1.20*** | 1.21*** | 1.21*** | 1.11 | 1.16* | 1.45*** |





| | | | | | | | |
|---|---|---|---|---|---|---|---|
| Risk perception (ref =no) | 0.79 | 0.81 | 0.8 | 0.78 | 0.79 | 0.79 | 0.78 |
| Experience of disaster (ref =no) | 1.35** | 1.37** | 1.36** | 1.35** | 1.37** | 1.35** | 1.34** |
| Homeownership (ref =rent) | 1.48*** | 1.49*** | 1.48*** | 1.48*** | 1.50*** | 1.49*** | 1.48*** |
| Employment (ref =no) | 0.74* | 0.73* | 0.74* | 0.74* | 0.73* | 0.74* | 0.74* |
| Number of adults | 1.08 | 1.07 | 1.07 | 1.08 | 1.07 | 1.08 | 1.08 |
| Lower Capacity ×Response efficacy | | 1.33** | | | | | |
| Caring for older/disabled adults×Response efficacy | | | 1.21* | | | | |
| Caring for Children×Response efficacy | | | | 0.95 | | | |
| Lower Capacity ×Self-efficacy | | | | | 1.22* | | |
| Caring for older/disabled adults×Self-efficacy | | | | | | 1.11 | |
| Caring for Children×Self-efficacy | | | | | | | 0.76** |
| **Model fit** | | | | | | | |
| LR Chi-square | 319.08 | 329.87 | 323.84 | 319.37 | 324.38 | 320.44 | 327.53 |
| Degree of freedom | 18 | 19 | 19 | 19 | 19 | 19 | 19 |
| p-value | | | | P<0.001 | | | |
| Pseudo R-squared | 10.03% | 10.36% | 10.18% | 10.04% | 10.19% | 10.07% | 10.29% |

[a] The reference categories are in parentheses

*p < .05, **p < .01, ***p < .001

**Table 6 Visual summary of statistically significant variables**

| Variable | Adequate preparedness | Minimal preparedness |
|---|---|---|
| Lower Capacity (ref =no)[a] | □ | ■ |
| Caring for older/disabled adults (ref =no) | ■ | ■ |
| Caring for Children (ref =no) | □ | ■ |
| Age (ref = 18–59) | □ | ■ |
| Gender (ref =male) | ■ | ■ |
| Education | □ | ■ |
| Race (ref =white) | □ | □ |
| Hispanic (ref =no) | □ | □ |
| English (ref =no) | □ | □ |
| Income (ref =less than $59,999) | ■ | □ |
| Disaster Information (ref =no) | ■ | ■ |
| Response efficacy | □ | □ |
| Self-efficacy | ■ | ■ |
| Risk perception (ref =no) | □ | □ |
| Experience of disaster (ref =no) | ■ | ■ |





| | | |
|---|:---:|:---:|
| Homeownership (ref =rent) | ■ | ■ |
| Employment (ref =no) | ■ | ■ |
| Number of adults | □ | □ |
| Lower Capacity ×Response efficacy | □ | ■ |
| Caring for older/disabled adults×Response efficacy | □ | ■ |
| Caring for Children×Response efficacy | □ | □ |
| Lower Capacity ×Self-efficacy | □ | ■ |
| Caring for older/disabled adults×Self-efficacy | □ | □ |
| Caring for Children×Self-efficacy | □ | ■ |

■=Statistically significant □ = Not statistically significant

Reference groups are in parentheses. See Tables 4–5 for odds ratios and p-values.

## 5 Discussion

This study employs a nationally representative data sample designed to examine the specific impact of factors affecting

family preparedness on the preparedness of American families, based on the concepts of minimum and adequate preparedness. Further, we investigated the effects of self-efficacy and response efficacy on the preparedness of vulnerable families.

Hypothesis 1 predicts that both response efficacy and self-efficacy are significantly correlated with disaster preparedness. The data partially support this hypothesis: as illustrated in Tables 4 and 5, self-efficacy is significantly

correlated with both adequate and minimal preparedness, indicating that higher self-efficacy is consistently associated with better levels of household disaster preparedness, which is consistent with previous studies (Adame and Miller, 2015; Adams et al., 2017). However, we found no statistically significant correlation between high response efficacy and adequate or minimal preparedness, and only families that considered preparedness useful were likely to be prepared. This differs from PMT & PADM and previous empirical studies (Bubeck et al., 2013; Chen and Cong, 2022; Grothmann and

Reusswig, 2006; Lindell and Perry, 2012; Tang and Feng, 2018). This indicates that although an individual or family recognizes that being prepared is effective, there could be other reasons that prevented them from adopting better-prepared actions. Similarly, public education or training can provide people with knowledge and information about preparation and inform them about its significance, but it does not necessarily lead to actively preparing (Miller et al., 2013; Paton, 2003).

Hypothesis 2 is also only partially supported. When considering minimal preparedness, households with LC, CC, and

COD do not report a higher probability of preparedness; contrary to many previous studies (Bronfman et al., 2019; Levac et al., 2012), they could reduce the likelihood of these households adopting disaster preparedness actions. A decline in care recipients' physical and cognitive functions could strengthen the barrier to disaster preparedness for caregivers. This is because for those being cared for, the process of disaster preparedness can be further complicated by issues such as medical equipment and the unique needs of daily life, given their limited mobility and the large number of medical services they

require (Bhalla et al., 2015; Christensen and Castañeda, 2014; Dostal, 2015).



Zamboni and Martin (2020) found that families with children have less disposable income and are less likely to prepare for resource-based programs. Notably, COD households are likely to report that they are adequately prepared. At first glance, this contradicts the concept of minimal preparedness, but on reflection, makes sense. Elderly/disabled adults may be ineffective in adopting specific disaster preparedness actions but may possess disaster experience or rich personal experience,

which would cause them to include basic supplies of food and water in their daily lives, besides reminding their caregivers or family members to take further steps to be better prepared for disasters (Shenk et al., 2009; Tomio et al., 2012). LC or CC families lacking the characteristics of COD families could be demotivated from being prepared and reduce the likelihood of them actively preparing for a disaster. This is perhaps related to the physical, psychological, social, and economic challenges that vulnerable groups present to their families (LaManna et al., 2020; Máximo et al., 2020). Such families require clear

instructions and guidance to help them make specific and complex preparations. Simultaneously, vulnerable families tend to spend much time on vulnerable groups, so they may not have adequate time to participate in training and drills related to disaster preparedness (Wakui et al., 2017). In addition, although LCs could be experienced in facing disasters or possess rich personal experience, their poor health conditions and lack of care services limits their capacity and ensures that they cannot prepare for disasters.

The data partially support Hypothesis 3. Although we were unable to demonstrate a significant correlation between response efficacy and disaster preparedness in the main effect, this did not affect the contribution of response efficacy to certain vulnerable families. Models 2–5 in Table 5 demonstrate that the moderating effect of response efficacy is only significant for LC and COD families but not for CC families. This outcome confirms the importance of response efficacy in promoting family disaster prevention and preparedness and reveals that response efficacy can have different moderator

effects for vulnerable families. In other words, the barriers to disaster preparedness for each household will vary according to the different vulnerable groups, and higher response efficacy can help overcome these barriers to disaster preparedness of LC and COD households but cannot effectively alleviate the barriers to disaster preparedness of CC households. However, the underlying cause is unclear and beyond the scope of this study. Future research can delve deeper into why response effectiveness does not alleviate barriers to preparedness in CC households through qualitative design.

Similarly, based on models 5–7 in Table 4, self-efficacy has no moderating effect on the disaster preparedness of COD households, even at a minimal level. This is attributable to COD groups being rich in experience and having family caregivers to assist them in adopting disaster preparedness measures. They could already possess relatively high self-efficacy, so self-efficacy is difficult to moderate their disaster preparedness. Self-efficacy is more critical in the context of LC and CC families, suggesting that higher self-efficacy helps these households overcome barriers to disaster preparedness, reduces the

gap between these and general households, and promotes minimal preparedness actions by vulnerable households. Our findings fully support the hypothesis of previous studies on LC that the respondents' reduced ability due to disability or health conditions is related to confidence in their ability to take action (Rao et al., 2022) and that lower self-efficacy could contribute to lower preparedness in this group (Eisenman et al., 2009; Marceron and Rohrbeck, 2019). Therefore, disaster



preparedness has been identified as an effective safety strategy for vulnerable groups, and educating them on how to plan for
disaster preparedness can largely contribute to better disaster preparedness among these groups (Zamboni and Martin, 2020).

Notably, neither response efficacy nor self-efficacy is significantly related to the adequate preparation of the three types
of families. In other words, although increased efficacy can increase the minimal level of disaster preparedness for
vulnerable families, neither response efficacy nor self-efficacy is a strong incentive for adequate preparedness. Is it the
family's vulnerability or the lack of other factors that prevent them from being better prepared, or do their limitations as a
vulnerable family prevent them from going beyond the level of minimal preparedness? The answers to these questions are
unclear, and future studies are necessary to further investigate this dynamic.

Among the control variables, older people over 60 are more likely to be prepared for disasters than younger people.
They could possess more experience with disasters, a better understanding of the risks involved, more resilience and strength,
and knowledge on how to prepare for disasters (Shenk et al., 2009). In addition, after their retirement, older people have
more time on their hands to communicate their disaster preparedness with doctors and others (Tomio et al., 2012).
Nevertheless, older people generally have poorer health and fewer financial resources and so are likely to be minimally
prepared rather than adequately prepared. Annual household income is not related to minimal preparedness, but households
reporting incomes above $60,000 are more likely to be adequately prepared, which calls for some low-cost disaster
preparedness measures so that families with weaker economic conditions can also be well prepared for a disaster. Women
are less likely to report being adequately prepared, but when it comes to minimal preparedness, women are more likely to
prepare with at least food, water, and transportation. This could be because women have more limited capacities in
emergencies, prompting them to prepare for disasters but causing them to be less likely to be adequately prepared due to
these limitations (Wakui et al., 2017). While this is consistent with the conclusions of previous studies (Kohn et al., 2012),
these differences could also arise from reporting biases (Hoffmann and Muttarak, 2017).

Homeowners suffer more losses than tenants during disasters, so they are more likely to take protective measures to
reduce losses. At the same time, owners have more opportunities to make structural changes to their houses, while tenants
are often not allowed to make structural changes (Grothmann and Reusswig, 2006). In addition, substantial evidence points
to renters being more likely to be low-income households forced to spend half of their income on rent and therefore have
limited funds available for disaster preparedness (Desmond, 2018). People with more education tend to be less likely to
prepare, perhaps due to their confidence in being self-sufficient. Furthermore, people with higher education levels may be
less likely to adopt preventive measures because they have more threat assessment barriers to their preparation (i.e., they
perceive a lower probability of risk and potential consequences) (Cong et al., 2021). As with multiple previous studies, we
found families with disaster experience to be more likely to be adequately prepared for a disaster (Kohn et al., 2012; Malmin,
2021; Rao et al., 2022). The studies by Malmin (2021) and Rao et al. (2022) have also reported significant associations
between previous disaster experiences and both adequate and minimal preparedness. Receiving information on disaster
preparedness during the past year effectively increased the likelihood of disaster preparedness within respondents'
households and highlighted the need for increased awareness and advocacy on what to do during disasters. Only 4.1% of our



sample reported not receiving information on disaster preparedness in the past year, but this area needs increased attention
and intervention. For example, during the early days of COVID-19, every household in Boston, Massachusetts, received
multilingual information packs designed to inform people of the risks and outline the resources available to them (Boston
delivers multilingual pamphlets on coronavirus to homes across city - The Boston Globe, 2022). Promoting similar messages
related to disaster preparedness is critical, especially among households with particularly vulnerable groups.

This study has several limitations. First, cross-sectional data are used in this study, so it would not be appropriate to
interpret any significant associations as being directional. Longitudinal studies would help address this limitation by
measuring the effectiveness of certain experiences or factors on the preparedness of specific populations or households over
time. Second, these data are self-reported, and while self-reporting is considered a reliable proxy for social cognitive
variables, it does not negate some reporting bias (Hoffmann and Muttarak, 2017). Specifically, respondents could experience
social pressure to appear more prepared than they are. For instance, self-reporting and optimism bias in responses could
result in respondents reporting higher efficacy and preparedness to sound more confident or lower efficacy to sound modest
(Rao et al., 2022).

**6 Conclusion**

This study directs attention to household preparedness among vulnerable populations, whom researchers have rarely
examined in previous disaster studies, and further examines the moderating role that response efficacy and self-efficacy play
between vulnerable households and disaster preparedness. The study findings highlight four main findings. Firstly, the
capacity of vulnerable households to adopt disaster preparedness actions is generally low. Secondly, improving response
efficiency can effectively increase the possibility of LC or COD families taking actions for minimal preparedness but does
not affect CC families. Thirdly, a significant correlation exists between self-efficacy and disaster preparedness actions of LC
or CC families, while no statistically significant correlation exists for COD families. Finally, neither response efficacy nor
self-efficacy was associated with the adequate preparedness of vulnerable households. These findings innovatively
emphasize the importance of response efficacy and self-efficacy in different family compositions. In other words, helping
vulnerable groups or families prepare for disasters makes them aware of the threat of disaster and, more importantly,
improves their self-efficacy and response efficacy. Based on these findings, targeted interventions and programs can be
designed to remedy the current lack of disaster preparedness education for vulnerable families and better promote practical
disaster preparedness activities for vulnerable families.

**Data availability**

Datasets related to this article can be found at https://www.fema.gov/about/openfema/data-sets/national-
household-survey.


**Declaration of competing interest**



The authors declare that they have no known competing financial interests or personal relationships that could have appeared to influence the work reported in this paper.

**Acknowledgement**

National Natural Science Foundation of China (Project No. 52008110).

Natural Science Foundation of Fujian Province (Project No. 2020J05195).

Innovative Methods Project of the Ministry of Science and Technology of the People's Republic of China (Project No. 2020IM010200).

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
