# Peer review of "The Role of Response Efficacy and Self-efficacy in Disaster Preparedness Actions for Vulnerable Households"

_EGUsphere, 2022_

## Referee Comment (RC1)

all in all, good research paper.

to revise:

mainly for the literature review chapter aim to give more details as some sections are not too clear.

Line 23

1.7 people around the world were ...

Line 124.

Change In contrast, to Additionally,

Line 180

Three Hypotheses not Four

Line 233

taking steps to prepare  help you get through a disaster in your area?" remove second to.

---

## Author Response (AR1)

**Response to Reviewer 1 Comments**

We sincerely thank you for thoroughly examining our manuscript and providing constructive comments to guide our revision. We have tried our best to revise the manuscript according to your kind and construction comments and suggestions. The responses to the comments are given below.

**Point 1:** Line 23 1.7 people around the world were ...

**Response 1:** We have accepted your suggestion and revised the statement of this sentence.

**Point 2:** Line 124. Change In contrast, to Additionally.

**Response 2:** We have accepted your suggestion to change In contrast to Additionally.

**Point 3:** Line 180 Three Hypotheses not Four.

**Response 3:** We have corrected the error.

**Point 4:** Line 233 taking steps to prepare to help you get through a disaster in your area?" remove second to.

**Response 4:** Thanks for your advice, we have cited your recommended and remove the second "to" in the Line 233.

**Response to Reviewer 2 Comments**

We sincerely thank you for thoroughly examining our manuscript and providing constructive comments to guide our revision. We have tried our best to revise the manuscript according to your kind and construction comments and suggestions. The responses to the comments are given below.

**Point 1:** Page 1, L 21 f.: "…and the possibility of climate and weather disasters has increased exponentially" – the possibility of disasters (how defined?), or the losses increased? Or is it the magnitudes and frequencies? – Please specify, as this is a bit too simple. It makes a difference in disaster preparedness if we focus on increasing frequencies or magnitudes, or both (and this is of course also hazard-dependent.

**Response 1:** Thank you for your suggestion. We have modified the expression of this sentence according to your suggestion. The revised part is as follows: "Global climate change has led to an exponential increase in the frequency and loss of natural disasters such as floods, droughts, tsunami, wildfires, thunderstorms, and hurricanes (Rao et al., 2022). According to a survey by the United Nations Office for Disaster Risk Reduction, in the ten years from 2005 to 2014 alone, about 1.7 billion people around the world were affected by various disasters, among which the Chinese people suffered the most disasters, while the United States sustained the most losses (United Nations Office for Disaster Risk Reduction, 2015)"

**Point 2:** Page 2, L 1 ff.: "…preparedness of households is one of the most effective ways of mitigating the impact of disasters (Gargano et al., 2015; Keim, 2008), and previous studies unmistakably confirm that the emergency preparedness of households significantly reduces the negative effects of disasters and ensures that people adequately support themselves and their families…" – other studies, however, conclude that there are some limitations and constraints, see as an example the works by Attems et al. (2020a; 2020b). – Please differentiate your statements here a bit. There are various trigger mechanisms that might (or not) encourage individual risk behaviour (Bamberg et al., 2017; Van Valkengoed and Steg, 2019).

**Response 2:** The expression of this sentence is indeed inappropriate. We have accepted your suggestion to modify this sentence and added relevant references. The revised part is as follows: "Several studies have indicated that the motivation behind individuals taking disaster preparedness measures is influenced by various factors, including socioeconomic factors, cognitive factors, individual experiences and knowledge. These factors intertwine with one another, resulting in a complex and uncertain nature of disaster preparedness motivation(Attems et al., 2020a, b). Some studies have shown that individual experiences, knowledge, and place attachment are positively associated with disaster preparedness behavior(Cong et al., 2014; Wang et al., 2021). Others have shown that these factors may not be significantly associated with disaster preparedness, or may be of lesser relevance(Bamberg et al., 2017; Van Valkengoed and Steg, 2019). It is worth noting that these studies generally emphasize the significant positive impact of response efficacy and self-efficacy on

promoting individual disaster preparedness behavior. This complexity highlights the multi-level nature of disaster preparedness motivation research and the importance of understanding the multiple factors behind individual behavior.
"

**Point 3:** Page 6, L 180: As already indicated by referee #1 there are only three hypotheses in this manuscript.

**Response 3:** We have corrected the error. The revised part is as follows: "Based on the above discussion, we developed three hypotheses; each hypothesis covers the three conceptualizations of preparedness"

Finally, we would like to thank you again for taking the time to review our manuscript, and we hope our correction can get your approval.